# REDCRAFT: A computational platform using residual dipolar coupling NMR data for determining structures of perdeuterated proteins in solution

Casey A. Cole[1]*, Nourhan S. Daigham[2], Gaohua Liu[3], Gaetano T. Montelione[2,4], Homayoun Valafar[1]

**1** Department of Computer Science & Engineering, University of South Carolina, Columbia, South Carolina, United States of America, **2** Department of Molecular Biology and Biochemistry, and Department of Biochemistry, Robert Wood Johnson Medical School, Rutgers, The State University of New Jersey, Piscataway, New Jersey, United States of America, **3** Nexomics Biosciences, Princeton, New Jersey, United States of America, **4** Department of Chemistry and Chemical Biology, and Center for Biotechnology and Interdisciplinary Studies, Rensselaer Polytechnic Institute, Troy, New York, United States of America

* coleca@cse.sc.edu

**Data Availability Statement:** All data files are available from the BMRB database (accession

## Abstract

Nuclear Magnetic Resonance (NMR) spectroscopy is one of the three primary experimental means of characterizing macromolecular structures, including protein structures. Structure determination by solution NMR spectroscopy has traditionally relied heavily on distance restraints derived from nuclear Overhauser effect (NOE) measurements. While structure determination of proteins from NOE-based restraints is well understood and broadly used, structure determination from Residual Dipolar Couplings (RDCs) is relatively less developed. Here, we describe the new features of the protein structure modeling program RED-CRAFT and focus on the new Adaptive Decimation (AD) feature. The AD plays a critical role in improving the robustness of REDCRAFT to missing or noisy data, while allowing structure determination of larger proteins from less data. In this report we demonstrate the successful application of REDCRAFT in structure determination of proteins ranging in size from 50 to 145 residues using experimentally collected data, and of larger proteins (145 to 573 residues) using simulated RDC data. In both cases, REDCRAFT uses only RDC data that can be collected from perdeuterated proteins. Finally, we compare the accuracy of structure determination from RDCs alone with traditional NOE-based methods for the structurally novel PF.2048.1 protein. The RDC-based structure of PF.2048.1 exhibited 1.0 Å BB-RMSD with respect to a high-quality NOE-based structure. Although optimal strategies would include using RDC data together with chemical shift, NOE, and other NMR data, these studies provide proof-of-principle for robust structure determination of largely-perdeuterated proteins from RDC data alone using REDCRAFT.

number 30494). https://bmrb.io/data_library/
summary/index.php?bmrbId=30494.

**Funding:** Funding for this project was provided by
National Institute of Health (https://www.nih.gov/)
grants P20 RR-016461 (to HV) and
1R01GM120574 (to GTM).The funders had no role
in study design, data collection and analysis,
decision to publish, or preparation of the
manuscript.

**Competing interests:** I have read the journal's
policy and the authors of this manuscript have the
following competing interests: G.L. is Chief
Scientific Officer at Nexomics. Biosciences, Inc. G.
T.M. is Scientific Founder of Nexomics
Biosciences, Inc. These affiliations are disclosed for
information purposes, and do not imply any bias in
the collection or interpretation of the data
presented here.

## Author summary

Residual Dipolar Couplings have the potential to improve the accuracy and reduce the
time needed to characterize protein structures. In addition, RDC data have been demon-
strated to concurrently elucidate structure of proteins, provide assignment of resonances,
and characterize the internal dynamics of proteins. Given all the advantages associated
with the study of proteins from RDC data, based on the statistics provided by the Protein
Databank (PDB), surprisingly only 124 proteins (out of nearly 150,000 proteins) have uti-
lized RDCs as part of their structure determination. Even a smaller subset of these proteins
(approximately 7) have utilized RDCs as the primary source of data for structure determi-
nation. One key factor in the use of RDCs is the challenging computational and analytical
aspects of this source of data. In this report, we demonstrate the success of the RED-
CRAFT software package in structure determination of proteins using RDC data that can
be collected from small and large proteins in a routine fashion. REDCRAFT accomplishes
the challenging task of structure determination from RDCs by introducing a unique
search and optimization technique that is both robust and computationally tractable.
Structure determination from routinely collectable RDC data using REDCRAFT can com-
plement existing methods to provide faster and more accurate studies of larger and more
complex protein structures by NMR spectroscopy in solution state.

This is a *PLOS Computational Biology* Methods paper.

## Introduction

Nuclear Magnetic Resonance Spectroscopy is a well-recognized and utilized approach to struc-
ture determination of macromolecules, including proteins. NMR spectroscopy has contributed
to structural characterization of nearly 12,000 protein structures deposited in the Protein Data-
Bank [1–3] (PDB). Although NMR studies may in general be more time consuming and costly
than X-ray crystallography, they provide the unique benefit of observing macromolecules in
solution conditions closer to their native environments and can provide information about
molecular interactions and internal dynamics at various timescales and resolutions.

Despite the changes that NMR spectroscopy has undergone over the years, the methodol-
ogy for analysis of NMR data has made relatively little progress. Nearly all methods of NMR
data analysis rely on a combination of Simulated Annealing [4,5], Gradient Descent [4,5], and/
or Monte Carlo sampling [4,5] to guide protein structure calculations in satisfying the experi-
mental constraints. The traditional approaches for characterizing protein structures by NMR
spectroscopy rely heavily on sidechain-sidechain based distance constraints [6], which are lim-
ited to interproton distances of 2.5–5 Å. The distance constraints obtained by NMR spectros-
copy are often augmented with other heterogenous data such as dihedral angle restraints based
on chemical shift data, scalar couplings, residual dipolar coupling (RDC), and/or paramagnetic
relaxation enhancement data. The structure of the target protein is then computed by deploy-
ing a combination of restrained Monte Carlo, molecular dynamics, and/or Gradient Descent
optimization routines. This combination of heterogeneous data and optimization techniques
with well documented limitations [4,7] has resulted in an inflated requirement for experimen-
tal data. The functional consequence of this process of protein structure determination has
manifested itself as inflated data acquisition time and cost of structure determination, while

also functionally limiting the upper boundary in the size of the proteins that can be studied by NMR spectroscopy.

RDCs are a promising source of data with unique strengths [8–14]. Generally, RDC data are more precise, easier to measure, and can provide informative structural and dynamic information. Because of their propensity to report on structure and internal dynamics of macromolecules, the utility of RDC data in structure determination can benefit from new approaches that operate in fundamentally different ways than those used by traditional software. These programs such as Xplor-NIH [15], CNS [16], and CYANA [17] have been modified to include RDCs in their calculations, but are not appropriate for *de novo* structure determination based on RDC data. Other contemporary methods have been presented [8,12,18–25] with a direct focus on characterization of structure from RDC data. While these programs address some of the shortcomings of the traditional approaches, their continued use of the conventional optimization techniques, such as Levenberg-Marquardt [26] or gradient descent, prevent full utilization of the rich information content of the RDC data. These approaches work for meticulously clean and complete datasets and therefore lack the robustness needed for the analysis of noisy or missing data. Some of these algorithms exhibit a direct or indirect reliance on completeness of the PDB archive, and therefore, rely on a thorough sampling of the protein fold-space [23,24]. Others utilize impractical numbers of RDCs [25,27–29] (e.g., 4 RDCs per residue collected in 5 alignment media) that cannot be routinely collected, especially on larger and perdeuterated proteins. Finally, there is no currently existing software that is capable of concurrent structure determination and identification of internal motion in proteins. REDCRAFT illustrates several unique advantages, with its most unique feature consisting of a novel search methodology optimally suited for the analysis of RDC data.

Here, as a proof of principle, we demonstrate the latest version of REDCRAFT that provides structure determination of proteins from Residual Dipolar Coupling (RDC) data that can be collected routinely for both small and large proteins. The most recent version of REDCRAFT (released in Dec. 2019 and available from: https://bitbucket.org/hvalafar/redcraft/src/master/) includes usability and methodological improvements [30]. In this report, we present the Adaptive Decimation feature that enables the use of less RDC data to study larger proteins. The impact of AD has been demonstrated recently in experiments using simulated data [30,31]. Here, we demonstrate the improved performance of REDCRAFT in application to experimental data. More specifically, we demonstrate the feasibility of structure determination of proteins using only RDCs that can be obtained from perdeuterated proteins, namely backbone N-C', N-H$^N$, and C'-H$^N$ RDCs in two alignment media. When available, our investigations are based on previously reported experimental RDC data, and when needed to provide appropriate test input data sets, these experimental data are augmented with synthetic data. We have demonstrated successful structure determination by REDCRAFT of eight proteins with a size range of 50 to 573 amino acids. Finally, REDCRAFT has been tested in RDC-only structure determination of a novel protein, PF2048.1, and the results were validated in comparison to conventional high-quality NOE and NOE plus RDC -based structures.

## Results

In the following sections we present three sets of results, all of which demonstrate structure determination of proteins from RDCs alone to reduce the overall cost of structure determination. The goal of these studies is not to promote an RDC-only strategy for protein NMR structure determination, but rather to demonstrate accurate structure determination with RDC-only data using REDCRAFT, with the aim of complementing these modeling calculations with other NMR data, where available. In the first set of results, we explored the structure

determination of proteins by REDCRAFT for which sufficient experimental RDC data were deposited into the BMRB database. In each of these exercises, we used substantially smaller set of RDC data than the previously reported comparable studies. In the second exercise, we first determined a high-quality structure of the novel protein PF2048.1 using traditional NOE-based methods, without and with RDC data. Using these two reference structures, we established the accuracy of the RDC-only structure from REDCRAFT, generated using only a fraction of the data. In the third set of results, we investigated the success of REDCRAFT in structure determination of larger proteins using synthetically generated RDCs. The structures of these proteins had been previously characterized by distance restraints including a small subset of RDCs, therefore establishing the plausibility of RDC collection for these proteins. To demonstrate the capabilities of REDCRAFT, our reported structures (except PF2048.1) have not been subjected to any energy refinement. While we have refrained from refinement steps in this work, clearly all structures can benefit from additional refinement using RDCs and any other experimental data.

REDCRAFT generates backbone conformations of proteins from backbone RDCs that is continuous along the protein sequence. Segments of residues for which RDC data are not available, or not properly interpreted because of internal dynamics, result in fragmentation of the available RDC data. In these cases, REDCRAFT supports structure determination of the backbone structure fragments where RDC data are contiguously available. While the RDC data does provide information about the relative orientations of these backbone structure fragments with respect to each other, as they are not sensitive to translation the precise positioning of these fragments with respect to one another can be determined by energetic constraints, or by additional NMR data.

## Protein structure calculation using experimental RDCs

Table 1 and Fig 1 summarize the results of REDCRAFT structure calculation of proteins using only experimental RDC data. The five proteins listed in this table have been previously studied by NMR spectroscopy, and experimental RDCs have been deposited in the BMRB [32]. As explained above, in some cases backbone RDC data are missing for a portion of a protein. In such cases, REDCRAFT accommodates fragmented structure determination and therefore, the structural comparison to the target structure is reported as a range representing the combined BB-RMSD's calculated for each fragment separately (columns 3 and 4). The fifth and six columns of Table 1 provide a quality of structural fitness to the RDC data as a Q-factor [33] reported for each of the alignment media separately. When structure calculation is conducted

**Table 1. Results for REDCRAFT's structure calculation using experimental RDCs.**

| Target Name | # Res. | BB-RMSD to NMR Structure[1] (Å) | BB-RMSD to X-Ray Structure[1] (Å) | Q Factor of Previously Solved Structure[2] | Q Factor of REDCRAFT Structure | % of Used Data |
|---|---|---|---|---|---|---|
| GB1 | 54 | 1.19 | 1.48 | 0.15, 0.11 | 0.07, 0.05 | 22% |
| GB3 | 54 | 1.9–2.5 | 1.3–2.2 | 0.17, 0.21 | 0.01–0.02, 0.02–0.29 | 23% |
| Rubredoxin | 50 | 1.12 | 1.02 | 0.15, 0.43 | 0.24, 0.13 | 41% |
| ChR145 | 145 | 1.4–2.3 | N/A | 0.17, 0.18, 0.24 | 0.01–0.05, 0.01–0.04, 0.01–0.03 | 11% |
| SR10 | 145 | 1.1–2.4 | 1.6–2.4 | 0.8, 0.68, 0.8 | 0.03–0.05, 0.05–0.16, 0.06–0.32 | 18% |

[1]PDB ids of reference NMR and X-ray structures are presented in Algorithms and Methods.

[2]Values separated by commas are for different alignment media; ranges are reported when the REDCRAFT structure determination results in multiple backbone fragments

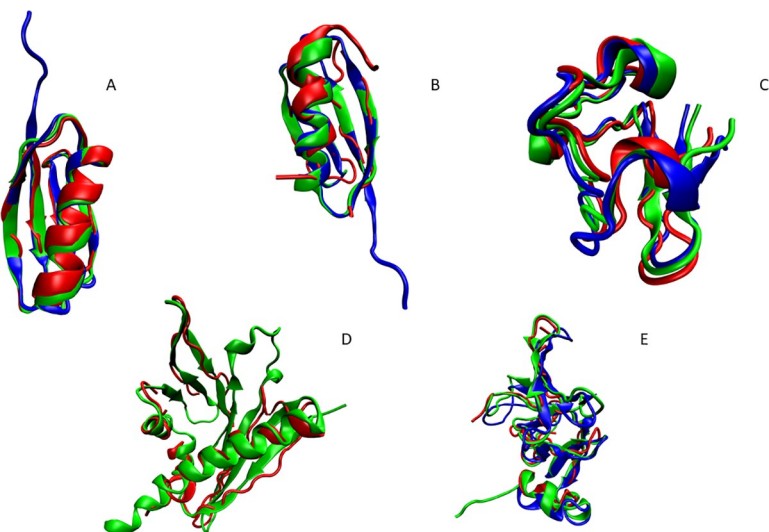

**Fig 1.** Results of REDCRAFT structure calculation (in red) compared to X-ray crystal structures (in blue) and, where applicable, traditional NMR structures (in green) for A) GB1, B) GB3, C) Rubredoxin, D) ChR145, and E) SR10.

in fragments (due to gaps in RDCs), the Q-factors reported for REDCRAFT structures (column 5) will consist of a list of ranges. Each item of the list (separated by a comma) reports the Q-factors for each alignment medium, while each range reports the minimum and maximum of Q-factors across all the fragments in a particular alignment medium. In summary, when using RDC data only, structures with Q-factors of over 0.5 are considered poorly fit structures, while values less than 0.3 are considered acceptable, and values between 0.3 and 0.5 indicate potentially acceptable structures. It is important to note that structures with higher Q-factors may correspond to an acceptable structure in the presence of additional experimental data. The last column of Table 1 indicates the percentage of the data that was utilized by RED-CRAFT compared to the number of constraints used previously. In general, as seen in Table 1, the obtained structures were less than 2 Å from the target structures with low Q-factors (indicating a reliable structure), while reducing the total data requirement by as much as 90% in some cases. In the following paragraphs, additional detailed results for each protein are discussed.

*GB1* –The previously calculated NMR structure of GB1 (2PLP) was determined using 769 RDC restraints that included N-H$^N$, N-C', C$_\alpha$-C', C$_\alpha$-H$_\alpha$, C'-H$^N$, C$_\alpha$-C$_\beta$ RDCs; 127 long range H$^N$-H$^N$ RDCs, and 54 Residual Chemical Shift (RCS) restraints from two alignment media [29]. In this study, 209 RDC restraints (compared to the total of 950 restraints) were used to obtain a structure with BB-RMSD less than 1.5 Å from both the X-ray and NMR structures. An example of the convergence of the top 50 ensemble structures resulting from REDCRAFT calculation for GB1 is shown in S2 Fig. The structures exhibit pairwise bb-rmsd of less than 0.5 Å from one another.

**GB3.** For GB3, the dataset included N-C', N-H$^N$, C$_\alpha$-H$_\alpha$, C$_\alpha$-C' RDCs, in five alignment media. Two previous studies used this full set of RDC data to determine a structure of GB3 with BB-RMSD within 1 Å of the corresponding X-ray structure [34,35]. In a previous RED-CRAFT study [36], the structure of GB3 was determined using N-H$^N$ and C$_\alpha$-H$_\alpha$ RDCs in two alignment media. Using these RDCs, REDCRAFT was able to reconstruct the structure to within 0.6–2.4 Å BB-RMSD of the high-quality NOE-base NMR structure. For the purposes of this study, the set of RDCs was reduced to contain just N-C' and N-H$^N$ RDCs, since these can

be collected using perdeuterated protein samples. Using these vectors, we were able to calculate a structure of this protein with BB-RMSD of less than 2.5 Å relative to the X-ray crystal structure.

**Rubredoxin.**    Previously, the structure of Rubredoxin was characterized to within 1.81 Å of the X-ray structure using N-C', N-H$^N$, C'-H$^N$, C$_\alpha$-H$_\alpha$, H$^N$-H$_\alpha$, H$_\alpha$-H$^N$ RDCs obtained in two alignment media.[10] Again, to simulate an RDC set that could be collected from a perdeuterated protein, this experimental RDC data set was reduced to N-H$^N$ and C'-H$^N$ RDCs only, from two alignment media. Using REDCRAFT, BB-RMSDs of 1.12 Å and 1.02 Å were obtained relative to the NMR and X-ray structures, respectively.

**ChR145.**    As part of the original study of ChR145, N-H$^N$ and N-C' RDCs were collected in two alignment media, and an additional set of N-H$^N$ RDCs were collected in a third alignment medium (PAG). All RDCs were deposited into the SPINE [37] database. Utilizing only these RDC restraints, REDCRAFT was able to produce structures with BB-RMSDs in the range of 1.4 Å—2.3 Å, relative to the traditional NMR structure that utilized 2,676 NOEs, 256 dihedral restraints and these same 328 RDCs.

**SR10.**    The structure of SR10 was obtained by NMR spectroscopy, with BB-RMSD of 2.0–2.5 Å with respect to the corresponding X-ray structure. The RDCs available for this protein were 3 sets of N-H$^N$ RDCs in three different alignment media. A fragmented study was utilized in this case due to large gaps in the RDC data. The original NOE-based structure utilized 1765 restraints (mix of RDCs and NOEs) whereas REDCRAFT only used only 320 RDC data.

## Conventional and REDCRAFT based structure determination of PF2048.1

Following conventional NOE-based structure determination procedures outlined in the Algorithms and Methods section, two ensembles of NMR-derived models of PF2048.1 were determined and deposited in the Protein Data Bank [38]. One structure ensemble was generated without any RDC data, using a total of 2,574 total restricting restraints, corresponding to 35.8 conformationally-restricting restraints per restrained residue (PDB_id 6E4J, BMRB_id 30494), and a second structure ensemble was generated that included RDC data (2,534 restricting restraints; 35.2 conformationally-restricting restraints per restrained residue) (PDB_id 6NS8 and the same BMRB_id 30494). In both cases, NOESY peak lists were assigned iteratively during the structure generation process (with or without RDC data); hence the sets of NOESY cross peak assignments and NOE-based restraints are slightly different between these two structures. Structure quality assessment metrics for these two NMR structures are presented in S1 Table (resulting structure in S1 Fig), and comparison of these two structures demonstrates the impact of RDCs in the structure determination. Overall, both structures (with and without RDCs) are high-quality structures, with excellent structure quality scores. The RDC Q-factors for the two alignment media M1 and M3 are 0.340 ± 0.020 and 0.320 ± 0.031, respectively for the models generated without RDCs, and 0.275 ± 0.015 and 0.280 ± 0.028, respectively, for models generated using RDC data as restraints. The DP scores [39], assessing how well the models fit to the unassigned NOESY peak list data, are 0.905 and 0.905 for the structures modeled without and with, respectively, RDC data. Molprobity packing scores [40], Richardson backbone dihedral angle analysis [40], and ProCheck [41] backbone and sidechain dihedral angle quality scores for well-defined regions of these models, are also excellent. The backbone RMSD between the medoid models [42] of the ensembles generated with and without RDC data is 0.745 Å. Taken together, this structure quality analysis demonstrates that both experimental NMR structures determined using conventional approaches are excellent quality, and good reference states for assessing modeling methods using RDC data alone.

The structure of PF2048.1 was also determined with REDCRAFT using only 228 RDCs, consisting of the backbone C'- H$^N$, N-H$^N$, N-C' RDCs from the first alignment medium (M1)

**Table 2. Results from structure calculation of PF2048.1 using 228 RDCs and secondary structure restraints are shown.**

|  | Q factor M1 | Q factor M2 | BB-RMSD to NOE structure w/o RDCs (Å) | BB-RMSD to NOE structure w/ RDCs (Å) |
|---|---|---|---|---|
| Before energy refinement | 0.04 | 0.16 | 2.38 | 2.24 |
| After energy refinement | 0.09 | 0.13 | 0.98 | 0.94 |

and backbone N-H$^N$ RDCs from the second alignment medium (M2). The final REDCRAFT structure exhibited BB-RMSD of 2.3 Å from the medoid NOE-only structure before any structural refinement. This structure was then subjected to 20,000 rounds of restrained energy minimization in Xplor-NIH, using the same 228 RDC restraints, in order to resolve some van der Waals collisions between secondary structural elements (helices 2 and 3). The Q-factors before and after minimization for both alignment media are shown in Table 2. The Q-factors for RDCs measured in alignment medium M1 incurred a slight increase during minimization due to the correction of van der Waals collisions in the computed structure. Fig 2 illustrates superimposition of the REDCRAFT computed structure of PF2048.1 (in red) before and after minimization, and the NOE structure without RDCs (in blue), or the NOE structure with RDCs (in yellow). The final structure exhibited Q-factors of 0.09 and 0.13 in the two alignment media respectively, and a BB-RMSD of less than 1.0 Å with respect to the representative (medoid) conformer of either of the NOE-based structures, determined with and without RDCs. An example of the convergence of the top 50 ensemble structures resulting from REDCRAFT calculation for PF2048.1 is shown in S3 Fig. The structures exhibit pairwise bb-rmsd of less than 1.005 Å from one another.

## Structure calculation of larger proteins

The results of structure calculation for larger proteins using synthetic RDCs are shown in Table 3 and Fig 3. Although the structure of ChR145 was determined by REDCRAFT using experimental data (reported in Table 1), here we have repeated the structure determination of this protein with synthetic data to illustrate the possibility of full structure determination (instead of a fragmented study) if adequate RDCs were collected. In this study, ChR145 was characterized in one full continuous segment with an overall BB-RMSD of 1.45 Å with respect to the reference structure. The resulting structure had excellent Q-factors.

In the cases of LpG1496 and Enzyme 1, fragmented studies were performed due to contribution of structural noise discussed in the Algorithms and Methods section. For instance, in

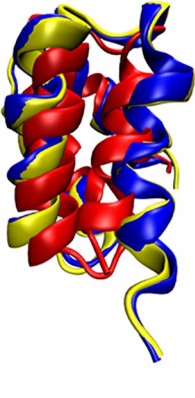
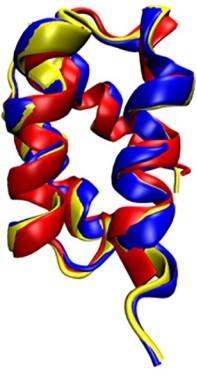

A                                                                    B

**Fig 2.** Results for PF2048.1 (in red) A) before energy minimization and B) after energy minimization are shown superimposed to the traditional NOE structure generated without RDCs (in blue) and with RDCs (in yellow).

**Table 3. Structure determinations of larger proteins by REDCRAFT using synthetic RDCs.**

| Target Name | Reference Structure | # Res. | BB-RMSD to Reference Structure (Å) | Q Factor of REDCRAFT Structure in M1, M2[1] |
|---|---|---|---|---|
| ChR145 | 2LEQ | 145 | 1.45 | 0.06, 0.05 |
| LpG1496 | 5T8C | 294 | 2.22 | 0.09, 0.07 |
| Enzyme 1 from E. *coli* | 2KX9 | 573 | 1.90 | 0.09, 0.07 |

[1]Values separated by commas are for different alignment media.

several cases, a single residue's dihedral angles were in severe violation of the Ramachandran space. In such instances, the structure determination was augmented with short refinement of each fragment followed by their integration using Xplor-NIH. For LpG1496, the largest contiguous fragment was 138 residues in length, displaying a BB-RMSD of 1.73 Å. Additional fragments ranged from 50 to 75 residues in length. All fragments reported Q-factors indicative of reliable structure in each alignment medium as well as low BB-RMSDs to the reference structure. The longest fragment for Enzyme 1 was 208 residues, which exhibited a BB-RMSD of 1.78 Å. All other fragments ranged from 50 to 100 residues in length. For the fragmented studies, all fragments were aligned to their respective structures and an average BB-RMSD was calculated (shown in Table 3).

## Discussion

RDCs report information on the overall tumbling, structure, and internal dynamics of a protein. Because of their convoluted information content, naïve analysis of RDCs could potentially produce faulty results. For instance, in the presence of dynamics, RDCs will be altered (due to averaging), and using them as restraints for static structure determination will produce an inaccurate structure.[43,44] However, more complete analyses of RDCs can provide a wealth of information including relative orientation of different domains or chains of a complex [45–48], calculation of structure [12,24,29,36,49,50] as demonstrated here, and information regarding internal dynamics [36,43,51–54].

Structure determination of protein with molecular weight greater than about 15 kDa by solution NMR spectroscopy is facilitated through perdeuteration of the sample protein, which suppresses nuclear relaxation pathways and provides sharper linewidths. Amide sites are reprotonated by back exchange, allowing [1]H-detected heteronuclear NMR studies, including some types of RDC measurements. However, the general absence of protons other than amide protons limits the kinds of RDC data that can be measured. The most pragmatic approach for the utility of RDCs in the study of such larger perdeuterated proteins necessitates the use of a limited set of RDCs collected in two alignment media. In this report we have demonstrated the

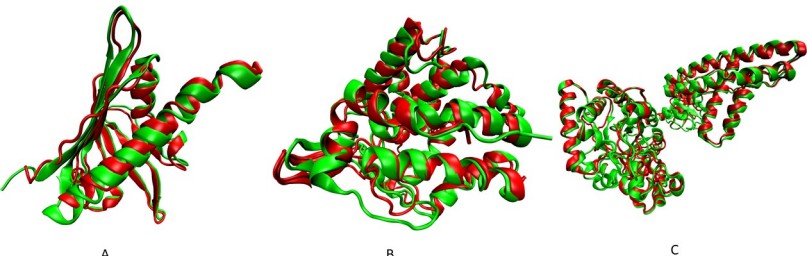

**Fig 3.** Results of REDCRAFT structure calculation (in red) compared to the reference structure (in green) A) ChR145, B) LpG1496 and C) Enzyme 1 from *E. coli*.

consistent success of REDCRAFT for structure determination of proteins with varying sizes (50–573 amino acid residues) using RDCs that can be collected from such perdeuterated proteins. In addition, assisted by the newly introduced Adaptive Decimation search, we have demonstrated reliable generation of multiple segments of protein structures with as little as 11–41% of the data used in previous RDC-based structure determinations. In this regard, Rubredoxin was an outlier in our studies because its previously reported structure was determined using an early version of REDCRAFT with an already reduced set of RDCs. The observed data reduction in the case of Rubredoxin provides additional support for the efficacy of Adaptive Decimation. Furthermore, we have shown that RDCs collected on perdeuterated proteins are sufficient for structure determination of large segments of larger proteins (as large as 573 residues) to accuracies of 1.5 to 2.2 Å BB-RMSD relative to the corresponding X-ray structures. This is a significant achievement since in most cases, such proteins must be perdeuterated to be amenable for study by solution NMR spectroscopy, and hence provide very sparse NOE data sets. Lastly, we have shown REDCRAFT's ability to characterize an unknown protein PF2048.1 with very little sequence or structural similarity to other characterized proteins. REDCRAFT was successful in structure determination of PF2048.1 (less than 2.5 Å BB-RMSD relative to the NOE-based structure, and < 1.0 Å BB-RMSD following restrained energy minimization) with as little as 228 RDCs (compared to > 2500 traditional restraints for the NOE-based structures).

In this report, we have stated the quality of raw structures produced by REDCRAFT without energy refinement to highlight its isolated ability in structure determination. However, under practical conditions, the raw structures produced by REDCRAFT should be subjected to restrained energy refinement. We have demonstrated the resulting improvement in the structural quality in the case of PF2048.1. This refinement step, using only RDC restraints, improved the structural similarity to structures determined with the full complement of restraints from 2.3 Å to less than 1.0 Å. The improvements in the refined structures result primarily from optimization of VDW interactions. Structural refinement with the conventional software also provides the opportunity of including the complete set of other available NMR data to remove some of the shortcomings of RDCs.

One feature of REDCRAFT is that gaps in RDC data along the backbone result in segments of modeled structures with limited information on how these segments are positioned with respect to each other. In particular, RDC data are insensitive to the relative translational positions of structural fragments. Under experimental conditions, gaps in the RDC data can be expected, in which case we have demonstrated REDCRAFT's ability to calculate structural fragments for those regions without such data gaps. Although RDC data from two alignment media can be used to orient two rigid structural segments with respect to each other [48], they cannot restrain the translational relationship between the two segments. Therefore, restrained energy minimization of the fragments resulting from REDCRAFT modeling and/or inclusion of complementary NMR restraints, are critical in finalizing the structure determination of the protein.

Structural elucidation of proteins from RDCs using REDCRAFT has other pragmatic features. For instance, characterization of protein structure does not have to be restricted to the entire protein. REDCRAFT's approach allows for structural investigation of a fragment of the protein as demonstrated with proteins GB3, ChR145, and SR10. Isolated study of a targeted fragment of a protein, or a segmentally labeled regions, can reduce the cost of structure determination and allow for study of larger proteins in a partitioned fashion. Furthermore, the combination of RDCs when analyzed with REDCRAFT, enables concurrent study of structure and dynamics of a protein as presented previously [36,43,44,49], also reducing the cost of such studies.

In conclusion, REDCRAFT can serve as a robust tool to analyze RDC data as an initial step in structure calculation using RDC data that can be obtained from perdeuterated proteins. Whether generating complete structure from a complete set of RDCs, or structure determination of fragments obtained from RDC data with gaps or segmentally labeled regions, REDCRAFT can produce structures within 2.5 Å of the native structure. Energy refinement of the REDCRAFT generated structure or structural fragments in the conventional software and/or addition of other experimental data can address the shortcomings of RDCs while reducing the overall needed experimental data. The ability to reliably obtain structures with less data (for instance using sparse NOE data sets obtained using perdeuterated proteins) provides a powerful method to study larger proteins by solution state NMR spectroscopy than cannot be addressed using conventional structure determination methods.

## Algorithms and methods

### Residual dipolar couplings

Residual Dipolar Couplings (RDCs) had been observed as early as 1963 in pneumatic solutions [55], and in the recent decades it has been pursued as an alternative approach to structure determination of macromolecules. This renewed interest in RDCs is based on advances in NMR spectroscopy [56] and the introduction of new media for anisotropic alignment of the samples [57–59]. RDCs have been shown to be valuable in structural characterization of proteins in solution [10, 14,60,61] and challenging proteins [44,62–66], while enabling simultaneous study of structure and dynamics of proteins [9,28,43,44,52,62,67,68].

The theoretical basis of RDC interaction [69–72] and their mathematical formulations [72, 73] have been extensively reported. Here, we directly focus on some of topics that relate to our discussion of REDCRAFT. In order to harness the computational synergy of RDC data, REDCRAFT utilizes the matrix formulation of this interaction as shown in Eq (1). The entity $S$ shown in Eq (1) and Eq (2) represents the Saupe order tensor matrix [46,55,69] (the 'order tensor') that is described as a 3×3 symmetric and traceless matrix. $D_{max}$ in Eq (1) is a nucleus-specific collection of constants, $r_{ij}$ is the separation distance between the two interacting nuclei $i$ and $j$ (in units of Å), and $v_{ij}$ denotes the corresponding normalized internuclear vector.

$$D_{ij} = \left(\frac{D_{max}}{r_{ij}^3}\right) v_{ij} * S * v_{ij}^T \qquad \text{Eq (1)}$$

$$S = \begin{bmatrix} s_{xx} & s_{xy} & s_{xz} \\ s_{xy} & s_{yy} & s_{yz} \\ s_{xz} & s_{yz} & s_{zz} \end{bmatrix} \qquad \text{Eq (2)}$$

$$v_{ij} = \begin{pmatrix} \cos(\theta_x) \\ \cos(\theta_y) \\ \cos(\theta_z) \end{pmatrix} \qquad \text{Eq (3)}$$

RDC data observed from any site on a protein will be influenced by the general alignment (anisotropic tumbling), structure, and the internal dynamics of the protein. Therefore, the proper use of RDCs must include a concurrent treatment of all three aspects of the data, which in turn increases the complexity of their analysis. An incomplete analysis of RDCs can have severe consequences such as generation of an inaccurate structure [36,43,44]. On the other hand, the proper detangling of the three components can provide information about the

alignment, structure, and the internal dynamics of the protein on biologically relevant time-scales [74–76] from a single source of data. In addition to the challenging nature of their analysis, RDCs impose an additional required step of creating a compatible alignment environment. Although RDCs impose this additional sample preparation step, their acquisition may be well justified in some instance when sufficient traditional NOE data are not available (e.g., for per-deuterated proteins).

Despite the advantages of RDCs for characterizing protein structures, only a handful of protein structures submitted to the PDB have been determined primarily by RDC data. The complexity of RDC analysis may lie at the core of the infrequent utilization of this rich source of data. It is therefore a useful exercise to fully understand the strengths of RDCs through the development of approaches that fully analyze the informational content of RDCs.

## REDCRAFT

Practically, all of the conventional NMR data analysis software packages such as Xplor-NIH [15], CNS [16], or CYANA [17] have been modified to incorporate RDC data into their analysis. Other specialized software packages [8,12,18–25] have contributed to the advancement of RDC data analysis in order to provide a more effective path to structure determination from RDC data. Although the contemporary approaches have been notably more successful in the recovery of structure from RDCs, they (conventional and contemporary approaches) have collectively been confounded by the challenges that are presented by the convoluted information content of RDCs.

To better illustrate the challenging nature of structure determination from RDCs (for simplicity assuming the absence of dynamics) and motivate the need for new approaches, we present Fig 4 that highlights the strengths and weaknesses of structure determination by RDCs and NOEs. Fig 4 represents the RDC and NOE fitness of 5000 decoy structures for PDB-ID 2KDI and 1GB1 (plotted on y-axis) as a function of their BB-RMSD to the known structure (shown on the x-axis). These 5000 decoy structures have been derived from the native structures by

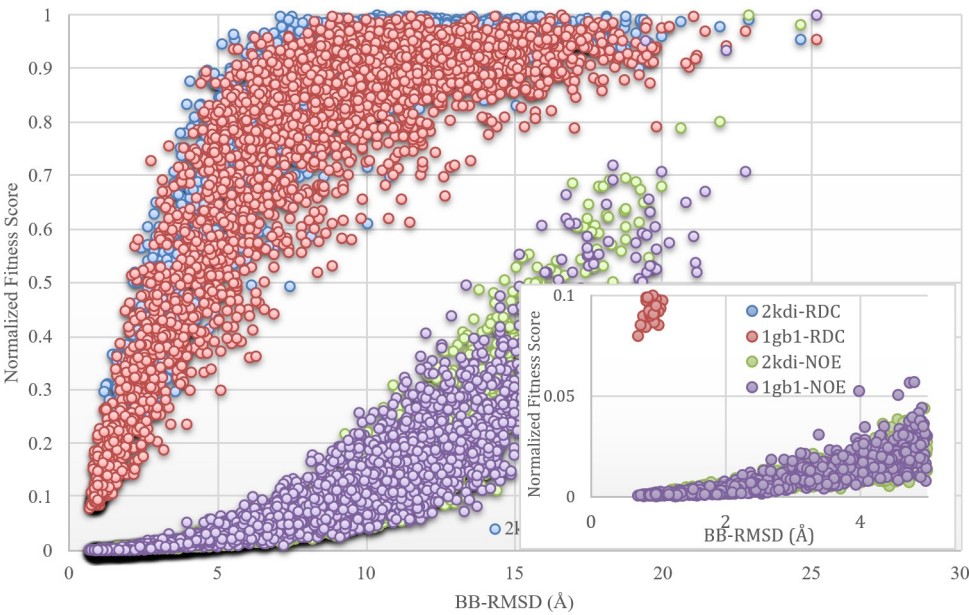

**Fig 4. RDC and NOE fitness of 5000 decoy structures generated randomly from a known structure versus their BB-RMSD to the actual structure.**

randomly altering the backbone torsion angles to achieve a continuum of distortions (measured in BB-RMSD's). The normalized fitness to both NOE and RDC experimental data (similar to Q-factor[31]) are calculated and plotted on the vertical axis. This figure illustrates the complementarity of NOEs and RDCs as reporters of protein structures. This figure suggests that NOEs are a relatively more sensitive reporters of structural fitness when the search is far from the native state. Therefore, NOEs are relatively more effective in guiding an extended initial structure toward the native structure. However, NOEs lose sensitivity to structural variation as they approach the native structure (as the curve flattens). Therefore, NOEs become relatively insensitive to structural variation when less than 2-3Å from the native structure. RDCs on the other hand exhibit a lack of sensitivity to structural variation when far from the native structure but gain sensitivity as they approach the native structure (steep incline and the tight scattering of the curve). The insensitivity of RDCs in structure determination from an extended structure is the primary cause of failure when using an extended structure as the starting point of search. The sensitivity of RDCs is generally the impetus in developing RDC based structure determination approaches. The existence of internal dynamics further complicates their use in structure determination endeavors.

**REDCRAFT's core engine.** REDCRAFT was previously introduced [12] to provide a more robust means of structure determination and detangling of structural information from internal dynamics [44,77]. The most recent version of REDCRAFT (released in Dec. 2019 and available from: https://bitbucket.org/hvalafar/redcraft/src/master/) aims to improve the usability of the software through the inclusion of a Graphical User Interface, compliance with NEF data exchange format [78], and inclusion of improved user documentation. Other software engineering design improvements allow for universal portability and usability in MAC, Linux, and Windows, and improved maintainability. In addition, the newest version of REDCRAFT integrates existing dihedral information either from experimental sources such as TALOS [79], or knowledge bases such as PDB [38] or PDBMine [80]. In this report, we present the Adaptive Decimation feature that has enabled the use of less RDC data to determine 3D structures of larger proteins. We will begin our discussion of REDCRAFT with some of the key features that enable unique analysis of RDCs and extend the discussion to introduce Adaptive Decimation.

One of REDCRAFT's unique features is its approach to structure determination that starts from a single amino acid, typically at the C or N termini of the sequence but can be anywhere in the sequence. This single amino acid is gradually elongated (in either direction) through the addition of one amino acid at a time to achieve a full-length protein. The flexible selection of a starting point of structure determination permits fragmented study of a protein. A fragmented study of a protein may be critical in the presence of gaps in the RDC data. The gradually increasing problem complexity has several computational and analytical advantages, as well as practical implications. The gradually increasing protein size allows REDCRAFT to avoid the pitfall of starting structure calculations from a fully extended structure (as outlined in Fig 4). The addition of one amino acid at a time allows for identification [43,44], and characterization of internal dynamics as illustrated elsewhere [43], therefore removing the effect of dynamics on structure.

**Adaptive decimation.** REDCRAFT requires a list of plausible torsion angles for each residue in the target protein. These lists can be automatically generated by REDCRAFT using different Ramachandran restraints, TALOS restraints, or knowledge bases such as PDBMine. An exhaustive combinatorial search of all possible torsion angles is clearly an intractable approach to the discovery of the globally optimal structure. REDCRAFT has incorporated several features such as Fixed Search Depth and Decimation to manage computational and space complexity of its search algorithm. During each round of elongation, a large number of structural

variants (in the order of 1,000,000 conformations) are evaluated for fitness to the RDC data. The Depth Search parameter selects only a fixed number of the fittest conformations (survival of the fittest, typically 1,000) to proceed to the next round of elongation. The presence of noisy RDC data may push the fitness of the globally optimal conformation to beneath the allotted depth search, at which time REDCRAFT will fail to produce the correct structure. The RED-CRAFT's search mechanism includes a Decimation feature [30,31,36] to allow representative members (selected based on clustering of structures) of the rejected conformations to proceed to the next round of elongation based on a static cutoff fitness to the RDC data. The Decimation feature is critical in improving the resilience of REDCRAFT to some quantity of erroneous or missing RDC data. While very effective, the proper selection of the cutoff threshold is critical in the successful recovery of the structure. In particular, when analyzing noisy data, selection of a high cutoff value for the decimation procedure is required, which either unnecessarily inflates the complexity of search at the early stages of structure calculation or loses effectiveness at the later stages of structure calculation. The Adaptive Decimation allows for an appropriately and automatically adjusted decimation threshold in order to remain effective at all stages of structure determination. A more effective decimation process will enable recovery from more erroneous or missing RDC data as demonstrated recently [30].

In this report we demonstrate the application of Adaptive Decimation in structure determination of proteins as large as 573 residues, with a more challenging set of RDC data that can be collected from perdeuterated proteins. When studying large proteins by NMR, it is a common practice to perdeuterate the sample in order to suppress $^1$H-$^1$H relaxation pathways and increase the transverse relaxation rates of the remaining protons, resulting in sharper line widths and improved signal-to-noise. Amide sites can be largely reprotonated by back exchange, resulting in a smaller selection of RDCs that can be measured. In particular, in the absence of sidechain protonation by biosynthetic methods, C'-H$^N$, N-H$^N$, and C'-N RDCs are the most easily obtained RDCs for perdeuterated proteins. It is therefore of great importance for any RDC-based structure determination technique to be able to characterize structures from this subset of data. In this report we demonstrate the success of REDCRAFT in protein structure determination using such sparse data types that are potentially available for perdeuterated proteins.

**REDCRAFT's general features.** REDCRAFT is developed using a sound Object Oriented (OO) programming paradigm, and it therefore lends itself well to encapsulation of the physical and biophysical properties of proteins. For instance, the construction of a Polypeptide object from the more fundamental Atom and AminoAcid objects, directly reflects the natural process of polymerization and translates into better source code readability as well as faster development and program execution. In addition, OO design allows for easier extendibility of the system. For example, while the main data source of REDCRAFT is currently RDCs, one could easily extend the architecture to use other orientational constraints such rCSAs [68]. The only changes that the developer would need to make is the scoring mechanism of the elongation process and addition of any new atoms needed for the new data source. Existence of the AminoAcid class makes the addition of new atoms straightforward.

REDCRAFT also provides several filtering and constraining tools that are uniquely useful for use with RDC data. For instance, Order Tensor Filter (OTF) allows selection of proteins based on prior knowledge of order tensors [81,82]. REDCRAFT also allows the user to define dihedral restraints. All restraints (including OTF and dihedral) can be flexibly turned on and off for select regions of a protein that may suffer from severe lack of experimental data. The most recent version of REDCRAFT (version 4.0) has also adopted NEF compliance in data import/export procedures [78], and has incorporated an advanced decimation process that has allowed for successful structure calculation of proteins with as much as ±4 Hz of experimental noise [30,31].

**Table 4. List of protein targets with their respective X-ray and NMR reference structures, RDCs used and the average BB-RMSD between the NMR and X-ray structures.**

| Target Name | NMR PDB ID | X-Ray PDB ID | No, Residues | RDCs in M1/M2/M3 | Avg. BB-RMSD |
|---|---|---|---|---|---|
| GB1 | 2PLP [29] | 1IGD [85] | 54 | [C'-H$^N$, N-H$^N$, N-C'] / [N-H] | 0.68Å |
| GB3 | 1P7E[77] | 1IGD [85] | 54 | [C'-H$^N$, N-H$^N$] / [C'-H$^N$, N-H$^N$] | 0.35Å |
| Rubredoxin | 1RWD [10] | 1IU5 [86] | 50 | [C'-H$^N$, N-H$^N$] / [C'-H$^N$, N-H$^N$] | 1.86Å |
| ChR145 | 2LEQ [37] | N/A | 145 | [N-H$^N$, N-C'] / [N-H$^N$, N-C'] / [N-H$^N$] | NA |
| SR10 | 2KZN [87] | 3E0O [88] | 145 | [N-H$^N$] / [N-H$^N$] / [N-H$^N$] | 2.91 |

## Evaluation

Our evaluation of REDCRAFT was conducted in three phases with increasing level of difficulty in structure determination. In the first phase, REDCRAFT was tested using a set of proteins with existing experimental RDCs and X-ray or NMR structures. In the second phase of the study, a novel protein was targeted for a simultaneous study by RDC (using REDCRAFT) and NOE-based structure calculation. In the last phase of the study, large proteins (larger than 500 residues) were chosen based on the availability of RDC data. Although a few large proteins have been subjected to RDC data acquisition, none contained enough RDC data to perform a meaningful structure calculation. In such instances, simulated RDCs were generated for a sparse set of interacting vectors. REDCRAFT was then used to calculate an RDC-based structure for each target protein to demonstrate the feasibility of RDC based structure calculation of larger proteins. The rationale for this phase is to illustrate the possibility of structure determination by RDCs when the collection of RDCs has been demonstrated in previous work. In each phase of the study structures calculated by REDCRAFT are compared to the existing NMR and X-ray structures (if applicable) of the respective proteins. The following sections provide more detailed information for each of the proteins as well as an overview of the RED-CRAFT algorithm.

## Target proteins

During the first phase of our experiment, we selected the target proteins (shown in Table 4) based on the availability of RDC data in BMRB or PDB, structural diversity, and existence of NMR or X-ray structure. RDC data for all the proteins except SR10 were obtained from the BMRB [32], while the RDC data for SR10 were obtained from the SPINE database [37]. Table 4 provides some self-explanatory information for each protein including the final column that highlights the average backbone similarity between the X-ray and NMR structures.

The protein GB1 has been previously studied in depth [29,83] and represents an ideal candidate to be used as a "proof of concept" case. GB3, an analog of GB1, was also investigated in this study using a different set of RDCs. The RDCs for the GB3 were previously collected [34, 35] for refinement of a solved crystal structure to obtain better fitness to experimental data (resulting in PDB ID 1P7E). Rubredoxin, represented another ideal target of study due to its mostly non-regular structure. Traditionally, structures that are heavily composed of helical regions prove difficult to solve for computational methods due to the near-parallel nature of their backbone N-H$^N$ RDC vectors. ChR145, represents a larger, mixed beta-sheet and alpha helix protein. In a previous study, this protein was extracted from the *Cytophaga hutchinsonii* bacteria and characterized using traditional NMR restraints (primarily NOEs). Of interest, ChR145's primary sequence is unique in the PDB. This fact alone makes its structural characterization difficult for any method that has a dependency on database lookups or homology modeling. SR10, a 145-residue protein, was characterized as part of the Protein Structure

**Table 5. List of protein targets used in the synthetic study of large proteins.**

| Target Name | X-ray PDB ID | NMR PDB ID | No. Residues | RDCs in M1/M2 |
|---|---|---|---|---|
| ChR145 | 2LEQ [37] | N/A | 145 | [C'-H$^N$, N-H$^N$, N-C']/[C'-H$^N$, N-H$^N$, N-C'] |
| LpG1496 | 5T8C | N/A | 294 | [C'-H$^N$, N-H$^N$, N-C']/[C'-H$^N$, N-H$^N$, N-C'] |
| Enzyme 1 from E. coli | N/A | 2KX9 [89] | 573 | [C'-H$^N$, N-H$^N$, N-C']/[C'-H$^N$, N-H$^N$, N-C'] |

Initiative [84] and was included in this study to represent a challenging case because of the low RDC data density. The RDC data for this protein consisted of only N-H$^N$ RDCs collected in three alignment media. Of additional interest, the RDCs were collected on a perdeuterated version of the SR10 protein.

Currently there are very few examples of larger proteins in the BMRB database that include a near complete set of RDC data from two or more alignment media. Where RDC data are available for large proteins, they are very sparse and generally available for only one alignment medium. Meaningful structure determination of proteins from RDC data requires RDC data in two or more alignment media [48]. Therefore, to investigate the feasibility of protein structure calculation of large proteins using only RDCs, synthetic sets of C'-H$^N$, N-H$^N$ and N-C' RDCs were generated in two alignment media using the software package REDCAT [45,46] as described previously [45]. A random error in the range of ±1 Hz was added to each vector to better simulate the experimental conditions. The proteins chosen for this controlled study are summarized in Table 5. Note that for ChR145 a synthetic study was also performed to demonstrate the unfragmented structure determination if additional RDC data had been acquired. It is noteworthy that Enzyme 1 from *E. coli* was chosen as an example of a large mixed α/β protein. The dataset used for solving the NMR structure of this protein included a very sparse set of N-H$^N$ RDCs that was not applicable in our studies but demonstrates the possibility of RDC data collection in large proteins.

In addition to the previously characterized proteins, RDC data were acquired for a novel, 71-residue protein (designated PF2048.1). PF2048.1 has been selected as a target of our studies due to its novelty in comparison to the existing archive of structurally characterized proteins. PF2048.1, an all-helical 9.16 kDa protein, exhibited less than 12% sequence identity to any structurally characterized protein in PDB (as of January 2019). The previously reported computational models of this structure [81] agreed on the helical nature of this protein and resulted in an ensemble of structures with as much as 10 Å of backbone diversity [81,82,90].

RDC data were acquired by NMR spectroscopy for this protein in Phage and stretched Poly Acrylamide Gel (PAG) alignment media. The resulting two sets of RDCs consisted of N-C', N-H$^N$ and C'-H$^N$ RDCs from the Phage and N-H$^N$ RDCs from the PAG media. The process of NMR data collection is described in Section 2.3. Collectively, the two data sets were missing ~17% of data points (48/276) leaving 228 total RDC data points (an average of 1.6 RDCs per residue, per alignment medium).

## PF2048.1

**Expression and purification.** Uniformly $^{13}$C,$^{15}$N-enriched PF2048.1, a 72-residue protein, was used for NMR structure determination, Prior to gene synthesis, the sequence was optimized by codon optimization software. The designed gene was synthesized by Synbio-Tech (www.Synbio-tech.com) and subcloned into pET21-NESG vector. Protein expression was performed by Nexomics Bioscience, as previously described [91]. Briefly, the recombinant

pET21-PF2048.1 plasmid was transformed into *E. coli* BL21 (DE3) cells and the cells were cultured in $^{13}C,^{15}N$-enriched MJ9 medium containing 100 μg/mL of ampicillin. The culture was further incubated at 37 °C and protein expression was induced by addition of isopropyl β-D-1-thiogalactopyranoside (IPTG) to the final concentration of 1 mM at logarithmic phase. Cells were harvested after overnight culture at 18 °C, cells were disrupted by sonication, and protein expression was evaluated by SDS-PAGE. The protein was purified using a standard Ni affinity followed by size exclusion two-step chromatography [91]. Since the purified PF2048.1 sample presented as 2 bands, an additional ion exchange chromatography was performed. The PF2048.1 sample from the two-step purification was pooled and dialyzed against buffer A (Buffer A: 20 mM Tris-HCl, pH 7.5), and loaded onto a HiTrap Q HP 5 ml column. A gradient of NaCl from 0 to 1 M was applied (Buffer B: 20 mM Tris-HCl, pH 7.5, 1 M NaCl). The PF2048.1 was pooled and concentrated to 1 mM using Amico Ultra-4 (Millipore). Protein samples were analyzed by SDS-PAGE (> 95% homogenous) and MALDI-TOF mass spectrometry.

**NMR sample preparation and data acquisition of PF2048.1.** NMR data were collected at 25 °C using Bruker Avance II 600 and 800 MHz spectrometers equipped with 5-mm cryoprobes. Sequence-specific backbone and side-chain NMR resonance assignments were determined using standard double- and triple-resonance NMR experiments, including 2D [$^1$H-$^{15}$N]-HSQC, 2D [$^1$H-$^{13}$C]-HSQC (aromatic region), 2D [$^1$H-$^{13}$C]-HSQC (aliphatic region), 3D HNCACB, 3D CBCAcoNH, 3D HNcoCA, 3D HNCA, 3D HNcaCO, 3D HNCO, 3D HBHA-coNH, 3D CCH-TOCSY, 3D $^{15}$N-edited TOCSY, and 3D simultaneous $^{13}$C-aromatic,$^{13}$C-aliphatic,$^{15}$N edited NOESY ($\tau_m$ = 80 ms). Processing of NMR spectra was done using TopSpin and NMRPipe [92], and visualization was done using NMRDraw and Sparky. NMR spectra were analyzed by consensus automated backbone assignment analysis using PINE [93] and AutoAssign [94] software, and then extended by manual analysis to complete the resonance assignments. The resonance assignments, together with raw fid data for all of these spectra and peak lists for the NOESY spectrum, are deposited in the BioMagResDatabase (BMRB ID 30494).

## Residual dipolar coupling measurements

For measurements under isotropic conditions a sample of $^{15}$N,$^{13}$C-enriched PF2048.1 was prepared at a concentration of 0.8 mM in 20 mM MES, 100 mM NaCl, and 5 mM CaCl$_2$ at pH 6.5. All samples also contained 10 mM DTT, 0.02% NaN$_3$, 1 mM DSS, and 10% D$_2$O. An anisotropic sample is required for the measurement of RDCs. After isotropic data collection, the PF2048.1 sample was used to prepare two partially aligned samples. A sample with pf1 phage as the alignment medium (designated alignment medium M1) was prepared which contained 0.88 mM PF2048.1 and 48 mg/mL phage in Tris buffer. After equilibration at room temperature for 10 min at 25°C the sample showed a deuterium splitting of 8.8 Hz when placed in the magnet. A second aligned sample was prepared in a 5 mm Shigemi tube using positively charged poly-acrylamide compressed gels (designated alignment medium M3). This sample contained approximately 0.77 mM PF2048.1. After equilibration at 4°C for 7–8 h the sample showed uniform swelling of the gel, which was then compressed vertically. Data were acquired for the isotropic and the two aligned samples to provide a complete set of $^{15}$N-$^1$HN, residual dipolar couplings. Data collection for the $^{15}$N IPAP-HSQC included 256 t1 points, and 2048 t2 points collected over 12 h. Residual dipolar couplings were calculated as the difference of the coupling measured in the aligned and isotropic conditions.

## Structure calculation with NOEs

Structures of PF2048.1 were determined from simultaneous $^{15}$N,$^{13}$C-resolved 3D-NOESY data, both with and without RDC data. In total 217 RDC measurements were used: 54 C'-H$^N$,

54 N-C', and 57 N-H$^N$ RDCs from medium 1 (M1—phage), and 52 N-H$^N$ RDCs from medium 2 (M3 –stretched polyacrylamide gel). Both ASDP [95] and CYANA3.97[17] were used to automatically assign long-range NOEs and to determine these structures. ASDP [95] was also used to guide the iterative cycles of noise/artifact NOESY peak removal, peak picking and NOE assignments, as described elsewhere [96]. NOE matching tolerances of 0.030, 0.03 and 0.40 ppm were used for indirect $^1$H, direct $^1$H, and heavy atom $^{13}$C/$^{15}$N dimensions, respectively, throughout the CYANA and ASDP calculations. This analysis provided > 2,300 NOE-derived conformationally-restraining distance restraints (S1 and S2 Tables). In addition, 132 backbone dihedral angle restraints were derived from chemical shifts, using the program TALOS_N[79], together with 70–74 hydrogen-bond restraints. Structure calculations were then carried out using ~35 conformational restraints per residue. One hundred random structures were generated and annealed using 10,000 steps. Similar results were obtained using both Cyana and ASDP automated analysis software programs. The 20 conformers with the lowest target function value from the CYANA calculations were then refined in an 'explicit water bath' using the program CNS and the PARAM19 force field [97], using the final NOE derived distance restraints, TALOS_N dihedral angle restraints, and hydrogen bond restraints derived from CYANA. Structure quality factors were assessed using the PDBStat [42] and PSVS 1.5[98] software packages. The global goodness-of-fit of the final structure ensemble with the NOESY peak list data were determined using the RPF analysis program. Structures determined with and without RDC data were deposited into the Protein Data Bank as entries 6NS8 and 6E4J, respectively.

## Structure calculations with REDCRAFT

REDCRAFT [12,30] was used to calculate the structures of proteins from RDC data with a standard depth search of 1000. Additional features such as Adaptive Decimation, minimization [36] (used parameters: 3, 1, 1-end, 5000), and 4-bond LJ [36] (threshold of 50) terms were included in all calculations. Although REDCRAFT is capable of including additional restraints such as Order Tensor Filter [36], dihedral restraints [80], we refrained from using these additional features. In particular, estimation of canonical order tensors in the absence of a structure [99,100] can be beneficial in this context, which we have not incorporated to highlight the computational capacity of REDCRAFT. For evaluation purposes, the RDC-RMSD reported by REDCRAFT was converted to Q-factor to assess the final models' fitness to RDC data using the software package REDCAT [45,46]. The backbone-RMSD (BB-RMSD) of REDCRAFT structures to existing structures were calculated using the *align* function of PyMOL [101] without the exclusion of any atoms. When comparing to NMR ensembles, RMSDs are computed relative to the representative (medoid) structure [42].

Under certain circumstances, RDC data may be absent for a segment of a protein. In the presence of gaps in the RDC data, REDCRAFT performs structure calculation of the protein in a segmented fashion. In such instances, the BB-RMSDs of the REDCRAFT fragments are reported as a range of minimum and maximum of the observed bb-rmsd over all the fragments. To highlight the success of REDCRAFT in structure determination, the raw structures calculated by REDCRAFT are reported for all proteins other than PF2048.1. In the case of PF2048.1, in addition to the raw structure calculated by REDCRAFT we performed restrained energy refinement. Restrained energy refinement is recommended in order to allow natural and allowable departure from ideal peptide geometries and resolve any existing backbone-backbone VDW violations. More specifically, we used XPLOR-NIH during the final refinement process, by subjecting the final structure to 30,000 steps of constrained Powell minimization that included the same set of RDCs used during the structure calculation with REDCRAFT.

## Supporting information

**S1 Table. Structure Calculation Input Files for PF2048.1.**
(DOCX)

**S2 Table. Structure Quality Statistics for PF2048.1.**
(DOCX)

**S1 Fig. Solution NMR structure of PF2048.1 refined with RDC (green) and without RDC (cyan).** The overall RMSD is within 0.6 Å.
(TIF)

**S2 Fig. Top 50 structures of GB1 reported by REDCRAFT.** The structural ensemble exhibit pairwise BB-RMSD of less than 0.5 Å.
(TIF)

**S3 Fig. Top 50 structures of PF.2048.1 reported by REDCRAFT.** The structural ensemble exhibit pairwise BB-RMSD of less than 1.005 Å.
(TIF)

## Author Contributions

**Conceptualization:** Casey A. Cole, Gaetano T. Montelione, Homayoun Valafar.

**Data curation:** Casey A. Cole, Nourhan S. Daigham.

**Formal analysis:** Casey A. Cole, Nourhan S. Daigham, Gaohua Liu.

**Funding acquisition:** Casey A. Cole, Gaetano T. Montelione, Homayoun Valafar.

**Investigation:** Casey A. Cole, Nourhan S. Daigham, Gaohua Liu.

**Methodology:** Casey A. Cole, Gaetano T. Montelione, Homayoun Valafar.

**Project administration:** Gaetano T. Montelione, Homayoun Valafar.

**Resources:** Gaetano T. Montelione, Homayoun Valafar.

**Software:** Casey A. Cole, Homayoun Valafar.

**Supervision:** Gaetano T. Montelione, Homayoun Valafar.

**Validation:** Casey A. Cole, Nourhan S. Daigham, Gaohua Liu.

**Visualization:** Casey A. Cole, Nourhan S. Daigham.

**Writing – original draft:** Casey A. Cole, Homayoun Valafar.

**Writing – review & editing:** Casey A. Cole, Nourhan S. Daigham, Gaohua Liu, Gaetano T. Montelione, Homayoun Valafar.

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
