## [Decision Letter · Decision Letter 0]

5 Aug 2020

Dear Dr. Cole,

Thank you very much for submitting your manuscript "REDCRAFT: A Computational Platform Using Residual Dipolar Coupling NMR Data for Determining Structures of Perdeuterated Proteins Without NOEs" for consideration at PLOS Computational Biology.

As with all papers reviewed by the journal, your manuscript was reviewed by members of the editorial board and by several independent reviewers. In light of the reviews (below this email), we would like to invite the resubmission of a significantly-revised version that takes into account the reviewers' comments.

We cannot make any decision about publication until we have seen the revised manuscript and your response to the reviewers' comments. Your revised manuscript is also likely to be sent to reviewers for further evaluation.

Sincerely,

Alexander MacKerell

Associate Editor

PLOS Computational Biology

Arne Elofsson

Deputy Editor

PLOS Computational Biology

Reviewer's Responses to Questions

**Comments to the Authors:**

Reviewer #1: The authors present an update to their REDCRAFT software, which is designed to calculate the structure of proteins from residual dipolar coupling data. Previous iterations of REDCRAFT have been described in several publications: J Magn Reson 2008, J Magn Reson 2010, J Biomol NMR 2014, and JCTC 2016. From what I could gather, this manuscript presents the largest benchmark of REDCRAFT to date, which is commendable. I think, however, that the manuscript could benefit from an extensive reorganization to highlight the novel features of REDCRAFT. This current version reads often as if the algorithm has never been described. In addition, I could not find a mention to exactly what new features the authors included in the latest version of REDCRAFT. The authors also fail to provide a URL for either the source code or the binaries of the software. This manuscript seems to be a benchmark and demo of REDCRAFT's abilities on a variety of proteins. While I find this important, new developments in NMR software should be encouraged, I cannot recommend the article for publication without major revisions. In particular, the authors should trim the contents of the manuscript to focus on a simpler message: what is new in REDCRAFT. Otherwise, the authors should just say, here's our code tested in a large benchmark and compared to traditional methods. Finally, I believe that there is room for some extra experiments (all computational) that could bring some novelty and added value to the work.

Below are my comments on each section of the manuscript that I hope can help the authors in their revisions:

(1) Abstract / Summary

- From the abstract and author summary, I cannot find what novel features have been added to this version of the software. The authors should make it very clear what is new in this version and why it matters to the users.

- As much as I like NMR, of the structures deposited in the RCSB PDB in the last 5 years, 50127 were determined by X-ray crystallography, 4468 by EM, and 2074 by NMR (both ssNMR and solution). I would change the very first sentence in the abstract to state that there are 'three primary experimental means of characterizing macromolecular structures'.

- I would also note that structure determination with/from RDC data is not recent. There are many papers from the early 2000s (so, nearly 20 years ago) describing applications and methods.

- On line 40, the authors write that structure determination with RDCs is limited to small proteins 'using RDC data from many alignment media (>3) that cannot be collected from large proteins'. Then, on line 45, they state that REDCRAFT can be applied to large proteins using simulated RDC data from perdeuterated proteins. So, can we use RDCs for large proteins or not?

(2) Introduction

- On line 72, if you are going to be as specific as '11.649', then do not use 'nearly'. I'd say 'nearly 12.000'.

- Regarding line 75, I would argue that NMR is not determining structures in their 'native' environment. The temperatures at which structures are determined and the concentrations used in the sample preparation are hardly comparable to 'native' conditions. Please remove the word 'native', as it induces the reader in error.

- On lines 78-80, I would add molecular dynamics simulations to the list of methods used by NMR software to produce structures. CNS and XPLOR-NIH both use MD for example. I would also argue that not having new methods isn't necessarily bad. Using old forcefields and not refining structures in explicit solvent, that I'd agree with that it's bad!

- On line 91, I disagree that the main limitation for studying large proteins with NMR is on the computational side, or with costs. Rather, in my experience, it's more to do with the spectral crowding and the inability to assign peaks productively.

- I need a citation for the sentence at lines 92-93: RDCs as a promising alternative to NOEs. By the way, there's a typo in Residual on that sentence.

- On line 107, the authors state that 'all the existing RDC-based structure determination approaches [...]', but wouldn't that include REDCRAFT, which was first published in 2008?

- On line 113, as in other parts of the manuscript, the authors refer to their search/optimization strategy as 'novel', but I have read the same exact strategy in their 2008 paper. Please remove 'novel'.

- On lines 116-117, the authors state the novelty of their work. Please make it easier to find! Do not bury the lede!

(3) Results

- On line 134, the authors make the assumption that because you can collect some RDCs for a (large) protein, you could collect more and thus pave the way to a RDC-only structure. I am not an expert in RDCs, but I would be very surprised if this is the case. Often, you can only get a few NOEs for a system.

- On line 145 the authors state that they calculate RMSDs as a range, when there are missing fragments of the protein. Why not simply present the RMSD over the existing atoms?

- When describing the data used for GB1 (lines 155 and below), the authors should state how many RDCs of each class they used in the final set of 209.

- On Table 1, the Q factors for some structures are quite high. For example, the REDCRAFT GB3 structure has a range of 0.02 to 0.34, which according to the authors' own definition, corresponds to saying 'extremely good to bad' fit. SR10 has Q factors of ~0.7 and above, so, very poorly fit. Could the authors explain, or maybe illustrate with a figure, which regions are well fitted to the data and which are not?

- On the same note, I would find it interesting to see the RMSD of the ensemble of final structures, as a measure of convergence of the structure determination process.

- On a more general note, the number of RDCs picked for each case seems quite arbitrary. If I was developing a method like REDCRAFT, I'd like to know what is the lower limit (ballpark) of data I need to use to get a good structure. As the authors state throughout the manuscript, researchers often use 'an excessive' amount of RDCs. It would be interesting to pick one of these cases (or all of them, they are not so many), and take a random sample of 50 RDCs, 100 RDCs, 150 RDCs, ..., and calculate when the structures start to converge to a fold. Maybe there is a relationship to be found between protein size/type and the optimal number of RDCs?

- The authors state on lines 205-206 that not using RDCs resulted in a Q factor of ~0.33, but using RDCs lowered it to ~0.28. Why such a small drop? I expected that optimizing the structure using RDCs and then comparing it with respect to how well it fits the data would yield much better Q factors. Can the authors comment on this?

- On line 220: 20.000 rounds of Powell? To resolve minor van der Waals (typo in the manuscript, please correct with appropriate capitalization) clashes? From experience, 50 steps of Powell are more than enough to clear clashes. Does XPLOR-NIH even get to the 20.000th step or it just converges before?

- Related to the structure determination of PF2048.1, a shift from 2.3 to less than 1.0A with minimization is quite substantial. I would wager that there is a lot that that EM step is doing in terms of optimizing the structure.

(4) Discussion

- On line 269: 'as little as 11% of the previously used data' is cherry-picking the lowest value. The average reduction is closer to 25%.

(5) Methods

- The section on RDCs is extremely length and unnecessarily detailed. Please summarize it to the bare minimum to understand the methods used in the manuscript and refer readers to reviews on the topic for more in-depth descriptions.

- On lines 346-347 the authors state that the energy landscape of RDC data is too complex to be navigated by GD/MC methods. However, if we refer to their own example of PF2048.1, it seems that simple EM was able to refine that structure quite substantially, using RDC data. I agree with the authors that specialized algorithms to handle RDC data are necessary and could probably be better than those written in the 80s (CNS/XPLOR), but the data presented in the manuscript does not back this statement.

- On line 352, the authors again refer to developments in 'recent years' and refer to REDCRAFT. Again, 2008 is 12 years ago, not necessarily recent. Please rephrase.

- Line 354: remove novel from the description of the search algorithm. The current manuscript does not describe any new methodology or feature, so I assume the two-stage method is the same as the original REDCRAFT paper, again, of 2008.

- On lines 369 and onwards, the authors describe how REDCRAFT is the only OOP software available to date. I hope they mean in the field of structural biology. OOP is only useful for developers and is quite tricky to get right. The authors rightfully state that one of the strengths of the paradigm is the ease of adding extensions to the original code. However, the examples they give (Polypeptide built from Atom and AminoAcid objects) are not novel nor unique to REDCRAFT. Biopython's PDB parser, written in 2003, has a similar hierarchical structure for its Structure representation (incidentally also using OOP). Without a very well-documented description of the API, the code base, the classes, their methods, it's nearly impossible for anyone to 'easily' extend REDCRAFT. Indeed, I see the authors have on their webpage a nice readthedocs page, but there is no description of the code/API. See the OpenMM documentation for a very good example of what you should aim for if you want to attract developers to extend your project.

Reviewer #2: uploaded as an attachment

**Have all data underlying the figures and results presented in the manuscript been provided?**

Reviewer #1: Yes

Reviewer #2: Yes

PLOS authors have the option to publish the peer review history of their article (what does this mean?). If published, this will include your full peer review and any attached files.

Reviewer #1: **Yes: **João Rodrigues

Reviewer #2: No
---

## [Decision Letter · Decision Letter 1]

25 Nov 2020

Dear Dr. Cole,

Thank you very much for submitting your manuscript "REDCRAFT: A Computational Platform Using Residual Dipolar Coupling NMR Data for Determining Structures of Perdeuterated Proteins in Solution" for consideration at PLOS Computational Biology. As with all papers reviewed by the journal, your manuscript was reviewed by members of the editorial board and by several independent reviewers. The reviewers appreciated the attention to an important topic. Based on the reviews, we are likely to accept this manuscript for publication, providing that you modify the manuscript according to the review recommendations.

Sincerely,

Alexander MacKerell

Associate Editor

PLOS Computational Biology

Arne Elofsson

Deputy Editor

PLOS Computational Biology

[LINK]

Reviewer's Responses to Questions

**Comments to the Authors:**

Reviewer #1: I would like to start by thanking the authors for their extensive and careful responses to my many previous comments. This revised version of the manuscript is substantially better and clearer for the reader. The methods are also more clearly presented and the results much better organized. I would only subdivide the REDCRAFT methods section to highlight the new algorithms. Nevertheless, I am happy to recommend this work for publication.

Reviewer #2: please see attached

**Have all data underlying the figures and results presented in the manuscript been provided?**

Reviewer #1: Yes

Reviewer #2: Yes

PLOS authors have the option to publish the peer review history of their article (what does this mean?). If published, this will include your full peer review and any attached files.

Reviewer #1: **Yes: **João Rodrigues

Reviewer #2: No
---

## [Editor Report · Decision Letter 2]

5 Jan 2021

Dear Dr. Cole,

We are pleased to inform you that your manuscript 'REDCRAFT: A Computational Platform Using Residual Dipolar Coupling NMR Data for Determining Structures of Perdeuterated Proteins in Solution' has been provisionally accepted for publication in PLOS Computational Biology.

Best regards,

Alexander MacKerell

Associate Editor

PLOS Computational Biology

Arne Elofsson

Deputy Editor

PLOS Computational Biology

---

## [Editor Report · Acceptance letter]

23 Jan 2021

PCOMPBIOL-D-20-01003R2 

REDCRAFT: A Computational Platform Using Residual Dipolar Coupling NMR Data for Determining Structures of Perdeuterated Proteins in Solution

Dear Dr Cole,

I am pleased to inform you that your manuscript has been formally accepted for publication in PLOS Computational Biology. Your manuscript is now with our production department and you will be notified of the publication date in due course.

With kind regards,

Alice Ellingham
